# Nickel/photoredox dual catalyzed arylalkylation of nonactivated alkenes

Yuxi Gao [1], Lijuan Gao[1], Endiao Zhu[1], Yunhong Yang[1], Mi Jie[1], Jiaqian Zhang[1], Zhiqiang Pan [1] ✉ & Chengfeng Xia [1] ✉

Alkene dicarbofunctionalization is an efficient strategy and operation-economic fashion for introducing complexity in molecules. A nickel/photo-redox dual catalyzed arylalkylation of nonactivated alkenes for the simultaneous construction of one $C(sp^3)-C(sp^3)$ bond and one $C(sp^3)-C(sp^2)$ bond has been developed. The mild catalytic method provided valuable indanethyla-mine derivatives with wide substrate scope and good functional group compatibility. An enantioselective dicarbofunctionalization was also achieved with pyridine-oxazoline as a ligand. The efficiency of metallaphotoredox dicarbo-functionalization was demonstrated for the concise synthesis of pharmaceutically active compounds.

The establishment of efficient protocols for the dicarbofunctionalization in one step is of sustaining passion in organic synthesis to improve the molecular complexity. Alkenes are abundant and ubiquitous motifs that are extensively utilized for conventional dicarbofunctionalization[1–10]. Along with the rapid development of visible-light-mediated photochemistry[11–26], the photoredox reagent has been applied as a synergetic catalyst to participate in the transition-metal catalytic cycle, giving access to emerging reaction manifolds[27–30]. The nickel/photoredox dual catalysis system has been developed as a powerful tool in the C − C bonds cross-coupling because of its high efficiency and mildness, as well as its green properties[31–34]. The nickel/photoredox catalytic dicarbofunctionalization was also successfully exploited to formulate two vicinal C − C bonds in one step, albeit mainly focused on electronically biased alkenes with directing groups or coordinating groups (Fig. 1a)[35–46]. In contrast, the nickel/photoredox catalytic dicarbofunctionalization of nonactivated alkenes presents a tremendous challenge because of their low reaction activities, giving rise to weak catalytic efficiency and more side reactions[38]. The sustainable development of the nickel/photoredox dual catalytic dicarbofunctionalization of non-activated alkene would enable creative approaches for the construction of valuable substrates. Till now, very limited examples were documented on the dicarbofunctionalization of nonactivated alkenes (Fig. 1b). Wu et al. developed a nickel/photoredox dual catalytic diarylation of ethylene[47]. Overman and co-workers exploited a

dual intramolecular dicarbofunctionalization of nonactivated alkenes from homoallylic oxalates catalyzed by nickel/photoredox[48].

We envisioned that a nickel/photoredox dual catalytic intramolecular arylalkylation of alkenes would provide a new avenue for the synthesis of valuable indanethylamine derivatives in one step (Fig. 1c). Yet, there could be potentially hindered by several parameters. The lower affinity and lower activity of nonactivated alkenes for nickel catalysis may hinder migratory insertion, leading to the reductive hydrogenation of halogen benzene or the direct cross-coupling of decarboxylative alkyl radicals with aryl halides[49]. Undesirable cyclization byproducts are often afforded in nickel-catalyzed intramolecular migratory insertion to the unactivated alkene[50,51]. Additionally, nickel catalyzed dimerization is also an alternative pathway[52–54]. Therefore, in the new synergistic catalytic design, the intramolecular migratory insertion to unactivated alkene should overwhelm the competitive dehalogenation pathway and the inter-molecular cross-coupling with an alkyl radical. Moreover, electro-philic alkyl radicals from photoredox decarboxylative α-amino acids should favor the oxidative addition with $Ni^{II}$-alkyl species to afford the $Ni^{III}$-dialkyl for the intermolecular alkyl−alkyl cross-coupling. Herein, we report the nickel/photoredox catalytic arylalkylation of nonactivated alkene via synergetic photoredox decarboxylation and nickel-catalyzed cross-coupling cyclization. Such a method would allow the regioselective construction of two vicinal C − C bonds at nonactivated alkenes, providing an efficient strategy for the rapid

---

[1]Key Laboratory of Medicinal Chemistry for Natural Resource, Ministry of Education, Yunnan Key Laboratory of Research and Development of Natural Products, School of Pharmacy, Yunnan University, Kunming 650500, China. ✉e-mail: panzhiqiang@ynu.edu.cn; xiacf@ynu.edu.cn

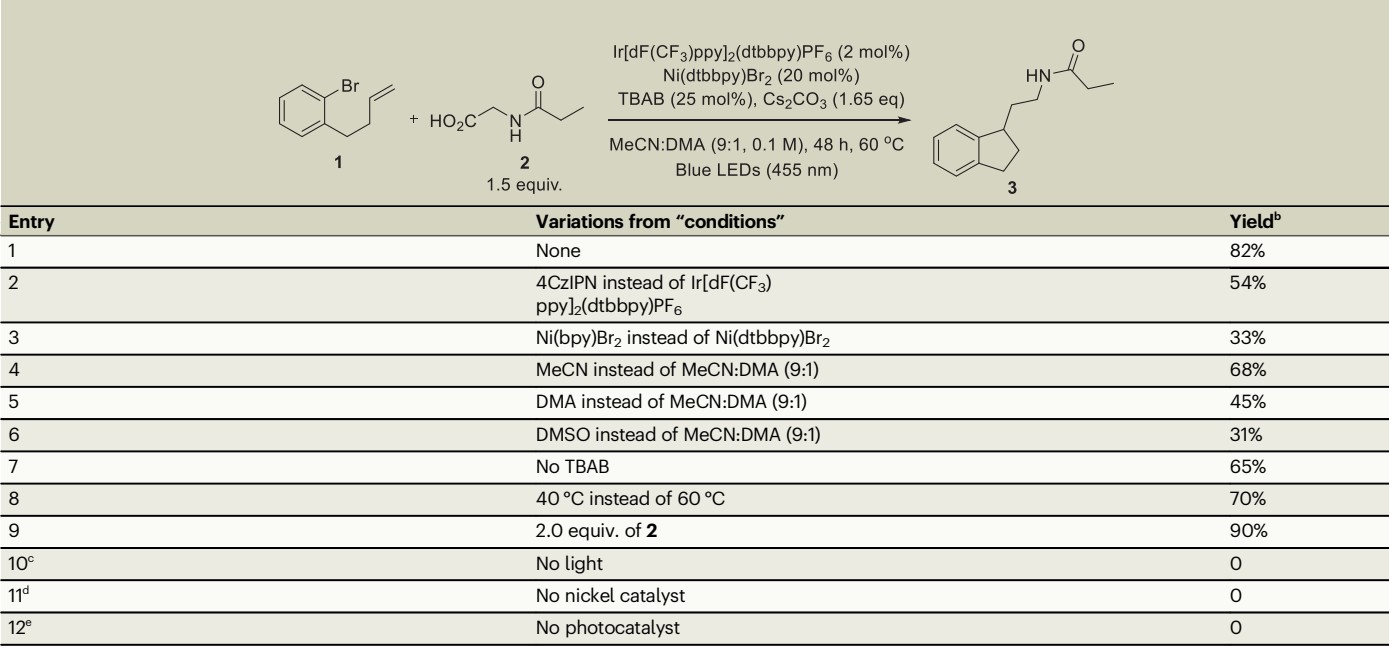

**Fig. 1 | Development of nickel/photoredox dual catalyzed arylalkylation of nonactivated alkenes. a** Nickel/photoredox catalytic dicarbofunctionalization of activated alkenes. **b** Nickel/photoredox catalytic dicarbofunctionalization of non-activated alkenes. **c** Metallaphotoredox catalyzed arylalkylation of nonactivated alkenes. 4CzIPN, 2,4,5,6-tetra(9H-carbazol-9-yl)−1,3-benzenedicarbonitrile.

construction of pharmaceuticals and pharmaceutically active compounds.

## Results and discussion

### Optimization of the reaction conditions

To begin, the butenylphenyl bromide **1** and the N-propionylglycine **2** were applied as modal substrates to probe the nickel/photoredox dual catalyzed intramolecular arylalkylation. After considerable optimization, the 1-indanethylamine **3** was afforded an 82% yield by using Ir[dF(CF$_3$)ppy]$_2$(dtbbpy)PF$_6$ as photoredox catalyst and Ni(dtbbpy)Br$_2$ as synergetic catalyst in the presence of TBAB and Cs$_2$CO$_3$ under irradiation with 18 W blue LEDs (455 nm) at 60 °C for 48 h (Table 1, entry 1). Other photocatalysts were not as so effective in the application of this metallaphotoredox catalysis, such as 4CzIPN afforded **3** in diminished yield (entry 2). Various nickel catalysts and ligands were carefully screened but were resulted in decreased yields too (entry 3). A mixture solvent (MeCN/DMA = 9:1) was more suitable than only MeCN, DMA, or DMSO (entries 4−6). The addition of TBAB to the reaction mixture apparently improved the efficiency (entry 7). When the reaction temperature declined to 40 °C, only 70% isolated yield was obtained (entry 8). Since the protodecarboxylation by-product, as well as other side-reactions, consumed the N-propionylglycine **2**[55,56], the optimal condition for photochemical arylalkylation was provided when 2.0 equivalents of **2** were involved (entry 9). Control experiments explained that the presence of nickel catalyst and photoredox catalyst under irradiation by visible light was significant for the reaction (entries 10−12).

### Scope of the reaction

With the optimized conditions in hands, we turned our attention to exploring the substrate scope of this transformation for nonactivated alkenes (Fig. 2). Various substituents, both

## Table 1 | Optimization of the reaction conditions[a]

| Entry | Variations from "conditions" | Yield[b] |
|---|---|---|
| 1 | None | 82% |
| 2 | 4CzIPN instead of Ir[dF(CF$_3$)ppy]$_2$(dtbbpy)PF$_6$ | 54% |
| 3 | Ni(bpy)Br$_2$ instead of Ni(dtbbpy)Br$_2$ | 33% |
| 4 | MeCN instead of MeCN:DMA (9:1) | 68% |
| 5 | DMA instead of MeCN:DMA (9:1) | 45% |
| 6 | DMSO instead of MeCN:DMA (9:1) | 31% |
| 7 | No TBAB | 65% |
| 8 | 40 °C instead of 60 °C | 70% |
| 9 | 2.0 equiv. of **2** | 90% |
| 10[c] | No light | 0 |
| 11[d] | No nickel catalyst | 0 |
| 12[e] | No photocatalyst | 0 |

*TBAB* tetrabutylammonium bromide, *4CzIPN* 2,4,5,6-tetra(9H-carbazol-9-yl)−1,3-benzenedicarbonitrile, *DMA* N,N-dimethylacetamide.
[a]Reactions were performed with **1** (0.2 mmol), **2** (0.3 mmol), photocatalyst (0.004 mmol), nickel catalyst (0.04 mmol), base (0.33 mmol), and additive (0.05 mmol) in 2.0 mL MeCN/DMA (9/1, V/V), were placed at approximately 8 cm away from two parallel LEDs (Blue LEDs, 455 nm, 18 W), and were heated at 60 °C in an oil bath for 48 h.
[b]Yield of isolated product.
[c]Without light.
[d]Without nickel catalyst.
[e]Without photocatalyst.

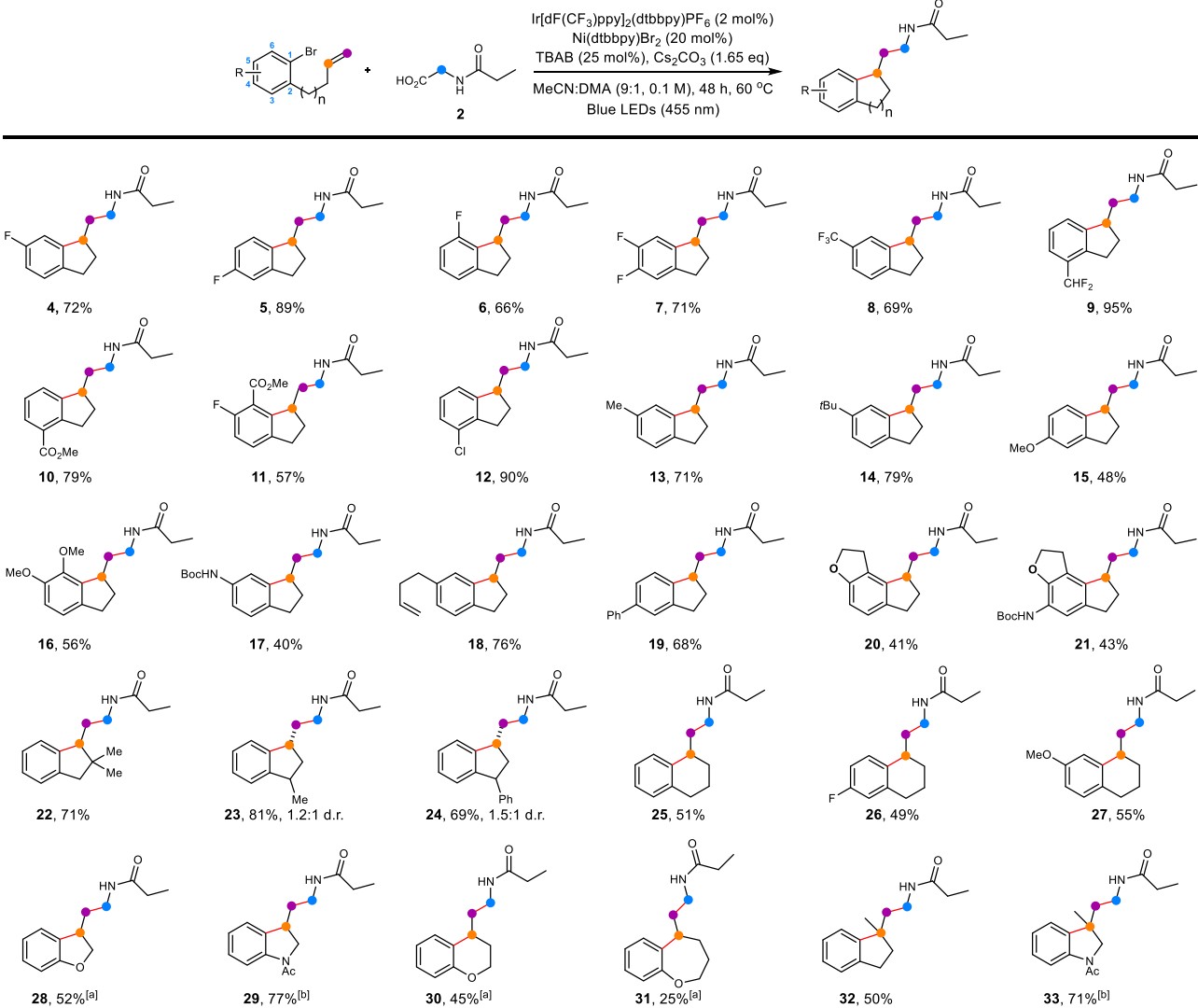

**Fig. 2 | Substrate scope for nonactivated alkenes.** [a] Reaction for 96 h. [b] Reaction for 24 h. TBAB tetrabutylammonium bromide, DMA *N,N*-dimethylacetamide.

electron-withdrawing (F, CF₃, CHF₂, CO₂Me, Cl) and electron-donating (Me, *t*Bu, OMe, NHBoc) were well tolerated in 3-, 4-, 5-, and 6-positions so that dicarbofunctionalized products were provided in moderate to excellence yields (4 – 17, 40%–95% yield). The electron-withdrawing substitution at the 3- and 4-positions of arene resulted in higher yields for dicarbofunctionalization (5, 9, 10, and 12). Additionally, excellent chemoselectivity for C($sp^2$)−Br bond cleavage over C($sp^2$)−Cl bond cleavage was observed in this process as demonstrated, affording 90% yield for compound 12. The protocol also tolerated the polysubstituted arene, and thus indanethylamine derivatives (7, 11, and 16) were afforded in synthetically useful yields (56%–71% yield). Note, that alkyl-substituted substrates (13, 14, and 18) generally provided better yields than alkoxy-substituted ones (15 and 16). The amide substituent was successfully suited for this process, installing the cyclic product in 40% yield (17). As shown for 18, the presence of an additional nonactivated olefin did not interfere with the dicarbofunctionalization. The formation of compound 18 as a single product suggested that the reaction was intiated from the nickel-catalyzed intermolecular cyclization instead of the radical addtion to alkene. Ramelteon (Rozerem™, 20) is a sleep agent which nearly has no adverse effects, such as drug dependence and cognitive impairment[57]. Our developed metallaphotoredox catalytic protocol proceeded smoothly to achieve the Ramelteon 20 and its derivative 21 from nonactivated alkenes in acceptable yields (41% and 43%,

respectively). Next, we examined the scope of nonactivated α-and β-substituted terminal alkenes (22 – 24). More sterically demanding di-α-substituted substrate was reactive, providing the desired product 22 in good yield (71%). As for mono-β-substituted substrates, diastereomeric ratios were observed under optimal reaction conditions (23 and 24). Delightedly, the current method was efficiently applied to the accurate construction of tetrahydronaphthalene derivatives with good yield for both electron-withdrawing and electron-donating substituents (25 – 27, 49%–55% yield). Besides the *C*-linked substrates, the *O*- and *N*-linked substrates were then subjected for the dicarbofunctionalization. It was found that the *N*-linked product (29) was harvested in much higher yield than the *O*-linked product (28). Meanwhile, we also tried to probe whether this dual-catalyzed strategy was applicable for generation of larger membered products. However, no 7-membered product was detected for the *C*-linked substrate. Instead, when *O*-linked substrates were exploited, the corresponding 7-membered cyclization compound 31 was isolated, ableit in low as 25% yield. Finally, the disubstituted terminal alkenes delivered products in moderate to good yields (32 and 33), while the internal alkenes failed.

We next set out to probe the substrate scope of α-amino acids for this method (Fig. 3). A variety of acyl groups were well tolerated in the indanethylamine formation (34 – 38, 43%–97% yield). In addition, *N*-cyclopropylcarbonylglycine was viable in the reaction (36),

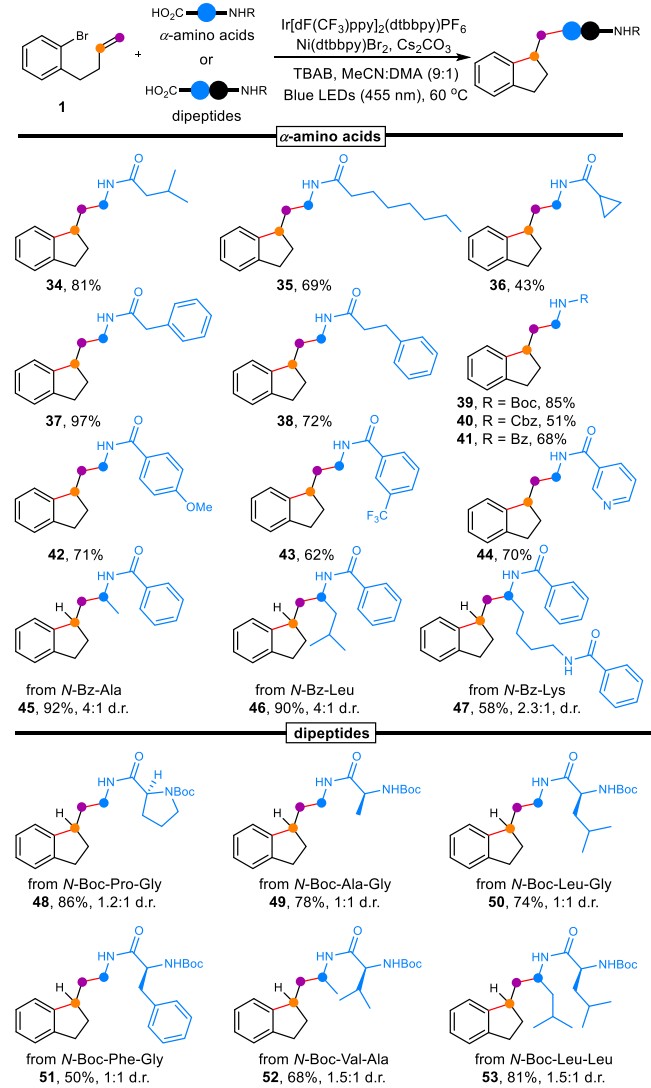

**Fig. 3 | Substrate scope for α-amino acids and dipeptides.** TBAB tetrabutylammonium bromide, DMA N,N-dimethylacetamide.

providing an acceptable yield (43% yield). The presence of Boc, Cbz, or Bz was also valid in the dual catalytic dicarbofunctionalization (**39** − **41**, 51%–85% yield). Various aryls, even N-nicotinoyl posed no challenge on the arylalkylation (**42** − **44**, 62%–71% yield). Several N-Bz-substituted natural α-amino acids were also examined. The reaction of nonpolar natural α-amino acids such as alanine and leucine proceeded smoothly to obtain the cyclic products in excellent yield (**45** and **46**, 92% and 90%). In terms of lysine, the yield of **47** was moderate (58%).

Encouraged by the above results, we wondered whether our strategy could be extended to apply dipeptides as substrates for the direct synthesis of indanethylamines. Delightedly, various glycine dipeptides could be employed as alkyl reagents for the visible-light photoredox/nickel catalytic arylalkylation of nonactivated alkenes, resulting in good yields (**48** − **51**, 50%–86% yield). As for N-Boc-Val-Al and N-Boc-Leu-Leu, corresponding indanethylamines were also afforded via the optimal conditions (**52** and **53**).

## Asymmetric dicarbofunctionalization

The asymmetric dicarbofunctionalizations of unactivated olefins have been documented in traditional transition-metal catalyzed cyclization, but an alkyl or an aryl group at the 2–position of terminal olefin was found to be necessary to improve the stereoselective migratory

insertion[52,54,58–61]. Fu and coworkers reported the only example of enantioselective synthesis of the tertiary stereogenic carbon with nonactivated alkenes via the nickel catalytic dicarbofunctionalization of the pre-prepared aryl boron substrates[62]. Since a tertiary stereogenic carbon was formed in this nickel/photoredox dual catalytic arylalkylation, we envisioned that an appropriate nickel catalyst and an efficient chiral ligand would realize the enantioselective synthesis of the tertiary stereocenter of indanethylamine via the stereoselective migratory insertion. The reaction parameters were re-optimized when chiral ligands were applied in this metallaphotoredox catalysis. After careful screening, the pyridine-oxazoline (Pyox) ligand[63–65], **L1**, was discovered as the optimal chiral ligand to afford (S)-**3** in moderate yield and high enantioselectivity (55% yield, 95:5 er. See ESI for the optimization of asymmetric reaction conditions). Note, that the standard photocatalyst, nickel catalyst, and additive have been revised to 4CzIPN, Ni(BF$_4$)$_2$.6H$_2$O, and MgCl$_2$ respectively (Fig. 4). With the optimized asymmetric condition in hands, an exploratory scope was implemented. A total of twelve compounds were illustrated in the asymmetric version with good enantioselectivities. In comparison to the electron-withdrawing substituents ((S)-**9** and (S)-**19**), the electron-donating substituents ((S)-**14** and (S)-**18**) proved to be more efficient in both yield and enantioselective protocol. More sterically indanethylamine (S)-**22** was also delivered in 46% yield and 96:4 er. Various acyl groups were well appropriate so that dicarbofunctionalized products were provided in good yields and high ee values ((S)-**3**, (S)-**37**, (S)-**39**, and (S)-**41**, 45%–85% yield, 93:7–96:4 er). Other amino acids (such as alanine, leucine and lysine) with additional subsituents on α-position were then evaluated for the enantioselective dicarbofunctionalizations. The tertiary stereogenic carbon displayed good enantioselectivity, while poor selectivity was observed on the amino acid moeity ((S)-**45** − (S)-**47**).

To demonstrate the utility of this nickel/photoredox dual catalyzed arylalkylation of nonactivated alkenes, applications for the concise synthesis of pharmaceutically active compounds were then conducted (Fig. 5). Melatonin receptor agonist ((S)-**55**) has high affinity and excellent selectivity for human MT$_1$ receptors, which nearly has no adverse effects such as drug dependence and cognitive impairment due to negligible affinity for MT$_3$ receptors[66]. The previous strategy for the synthesis of (S)-**55** required four steps from commercially available 6-methoxyl-1H-indanone[57]. With our developed asymmetric nickel/photoredox dual catalytic arylalkylation, the compound (S)-**55** was achieved in just one step from the known nonactivated alkene **54**, demonstrating the high efficiency of this protocol (Fig. 5a). The absolute configuration of the asymmetric indanethylamine (S)-**55** was determined via the careful contrast of the reported enantiomeric excess[66]. Another example illustrating the efficiency of this protocol was the S20242 (**57**), an Agomelatine derivative for the re-entrainment of sleep-wake cycles and the restoration of the body's core temperature rhythms[67]. The commercially available benzylbromide **56** reacted with 3-butenyl magnesium bromide under the catalysis of CuI to yield 2-bromo-4-methoxy-1-(pent-4-en-1-yl)benzene, which was then applied to the nickel/photoredox dual catalytic conditions with N-propionylglycine **2**. After oxidation with DDQ, the S20242 (**57**) was furnished in a three-step sequence (Fig. 5b).

## Mechanistic study

To shed light on the mechanism of this dual catalytic cycle, a series of mechanistic experiments were conducted (Fig. 6). Cyclic voltammetry (CV) studies of the deprotonation of N-propionylglycine **2** ($E_{p/2}^{red}$ = +0.98 V versus SCE in MeCN) (Fig. 6a) suggested that it can be oxidized by Ir[dF(CF$_3$)ppy]$_2$(dtbbpy)PF$_6$ ($E_{1/2}^{red}$ [*Ir$^{III}$/Ir$^{II}$] = +1.21 V versus SCE in MeCN)[68]. The Stern-Volmer quenching experiments revealed that the excited state of Ir[dF(CF$_3$)ppy]$_2$(dtbbpy)PF$_6$ was efficiently quenched by the anion of **2** (Fig. 6b). In contrast, no excited state quenching was observed for butenylphenyl bromide **1** (Fig. 6c).

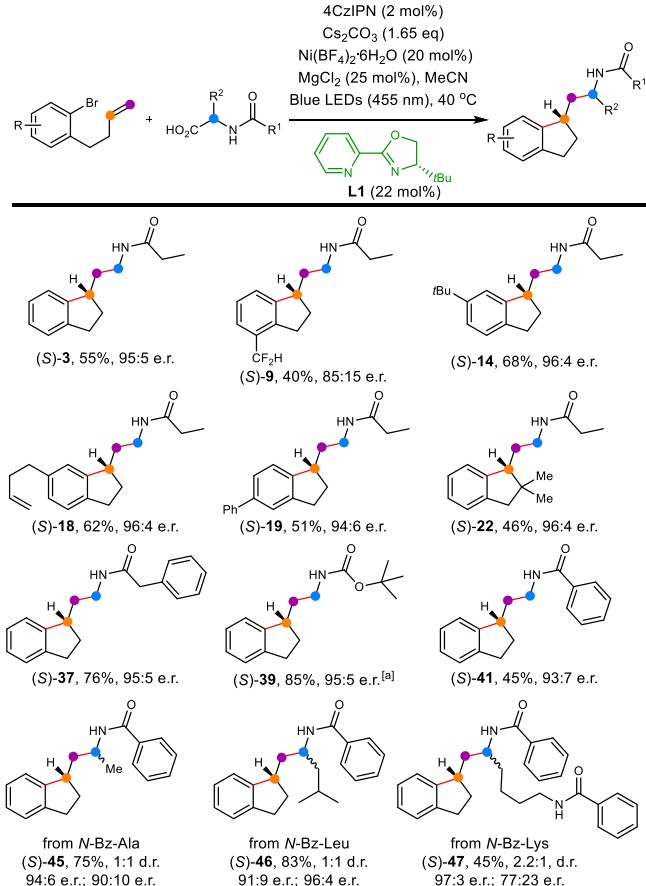

**Fig. 4 | Application in enantioselective arylalkylation of nonactivated alkenes.** Reaction conditions: nonactivated alkenes (0.20 mmol), $\alpha$-amino acids (0.40 mmol), photocatalyst (0.004 mmol), nickel catalyst (0.04 mmol), **L1** (0.044 mmol), base (0.33 mmol), and MgCl$_2$ (0.05 mmol) in 2.0 mL MeCN, were placed at approximately 8.0 cm away from two parallel LEDs (Blue LEDs, 455 nm, 18 W), and were heated at 40 °C in an oil bath for 48 h. [a] Without MgCl$_2$. 4CzIPN, 2,4,5,6-tetra(9*H*-carbazol-9-yl)−1,3-benzenedicarbonitrile.

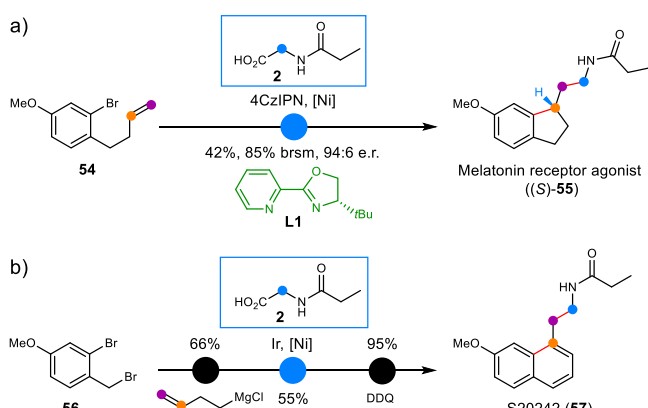

**Fig. 5 | Application in the concise synthesis of pharmaceutically active compounds. a** Synthesis of melatonin receptor agonist (S)-55. **b** Synthesis of S20242 (57).4CzIPN 2,4,5,6-tetra(9*H*-carbazol-9-yl)−1,3-benzenedicarbonitrile, DDQ 2,3-dichloro-5,6-dicyano-1,4-benzoquinone.

These results provided evidence that the deprotonated *N*-propionylglycine **2** was oxidized by the excited photocatalyst via single electron transfer (SET) oxidation followed by a decarboxylative process to afford an alkyl radical. The addition of (2,2,6,6-tetramethylpiperidin-1-yl)oxyl (TEMPO) or 1,4-dinitrobenzene as radical

inhibitor strongly inhibited the photoredox decarboxylation, indicating that a radical process was highly possible. A light-on-off experiment was carried out to verify that the reaction underwent a photochemical pathway (See Supplementary Figs. 8 and 9)[69].

To exclude the addition of olefins by free radicals, generated from amino acid decarboxylation, and then cyclization with Ni[II], two control experiments were conducted (Fig. 7). The reaction of **58** with **2** gave no cross-product, suggesting that a radical addition of the unactivated alkene is not favored. The selective coupling of radical with the two alkenes in compound **18** also confirms the reaction sequence (Fig. 2). Moreover, in the presence of 1,4-cyclohexadiene, the reaction of **S19** with **2** gave both the desired product **19** and protonated byproduct **60**[34,54]. These results support the reaction pathway that involves the activation of Ar−Br with Ni[0], intramolecular migratory insertion, and alkyl radical coupling.

Based on the above mechanistic studies and references, a plausible dual catalytic mechanism was proposed as shown in Fig. 8. The excited state Ir[III]* **A** oxidizes the deprotonated *N*-propionylglycine **2** via a SET process to afford the corresponding Ir[II] intermediate **B** and a carboxyl radical[68], which then delivers the alkyl radical **C** upon rapid release of CO$_2$. Concurrently with the photoredox cycle, the oxidative addition of the Ni[0]L$_n$ **D** into aryl bromide **1** provides an aryl-Ni[II] intermediate **E**, which undergoes an intramolecular $\beta$-migratory insertion of the nonactivated alkene and then a cyclization to afford the alkyl-Ni[II] intermediate **F**. Next, the addition of alkyl radical **C** to the Ni[II] species **F** generates the alkyl-Ni[III]-alkyl intermediate **G**, affording the dicarbofunctionalization product **3** and the Ni[I]L$_n$ species **H**. A SET event between the Ni[I] intermediate **H** ($E_{p/2}^{red}$ [Ni[I]/Ni[0]] = −1.29 V versus Ag/AgCl in MeCN, Fig. 6d)[34,70–72] and the Ir[II] intermediate **B** ($E_{1/2}^{red}$ [Ir[III]/Ir[II]] = −1.30 V versus Ag/AgCl in MeCN, Fig. 6d)[68] simultaneously regenerates the Ir[III] photoredox catalyst and the Nickel catalyst, thereby closing both catalytic cycles.

In summary, an arylalkylation of nonactivated alkenes enabled by photoredox/nickel dual catalysis had been developed. The metalla-photoredox dicarbofunctionalization of the nonactivated alkenes with $\alpha$-amino acids and butenylphenyl bromides resulted in the efficient synthesis of indanethylamine derivatives. This mild catalytic protocol displayed a broad substrate scope and a good functional group tolerance. An enantioselective strategy was then exploited to install the tertiary stereocenter with good yields and high enantioselectivities by using Pyox as ligand. This method was also demonstrated for the concise synthesis of pharmaceutically active compounds.

## Methods
### General procedure for nickel/photoredox dual catalyzed arylalkylation
To a 10 mL glass tube equipped with a septum and a magnetic stir bar was added Ni(dtbbpy)Br$_2$ (19.5 mg, 0.04 mmol, 20 mol%), Amino acid derivatives (0.40 mmol, 2.0 equiv.), TBAB (16.1 mg, 0.05 mmol, 25 mol%), Cs$_2$CO$_3$ (108 mg, 0.33 mmol, 1.65 equiv.), Ir[dF(CF$_3$) ppy]$_2$(dtbbpy)PF$_6$ (4.5 mg, 0.004 mmol, 2.0 mol%) and MeCN (1.8 mL) and DMA (0.2 mL) in the glove box. The corresponding unactivated alkenes (0.20 mmol, 1.0 equiv.) was added to the glass tube with a pipette gun under the argon. The resulting mixture was then sealed and wrapped with electrical tape and then irradiated with two parallel 18 W LEDs (455 nm,) from a distance of approximate 8 cm for 48 h. The reaction was maintained at 60 °C by heating in an oil bath and cooling by a fan. Then, the solvent was evaporated and concentrated, the residue was purified by silica chromatography.

### General procedure for asymmetric arylalkylation
To a 10 mL glass tube equipped with a septum and a magnetic stir bar was added Ni(BF$_4$)$_2$·6H$_2$O (13.6 mg, 0.04 mmol, 20 mol%), Ligand (9.0 mg, 0.044 mmol, 22 mol%) and MeCN (2.0 mL) in the

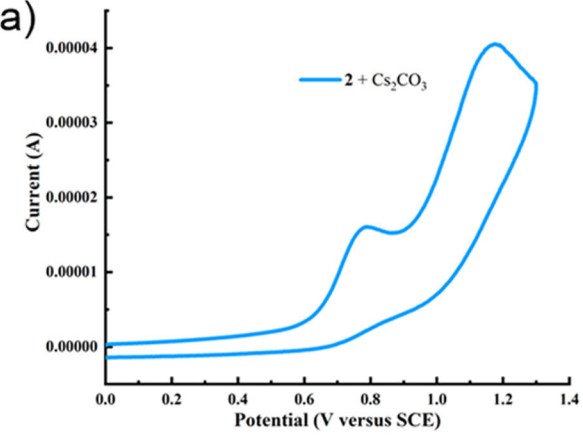

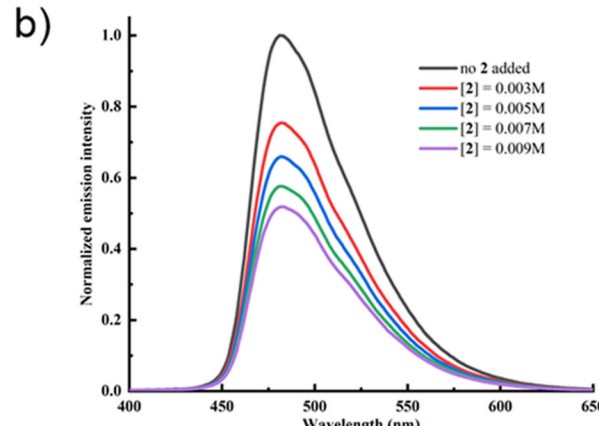

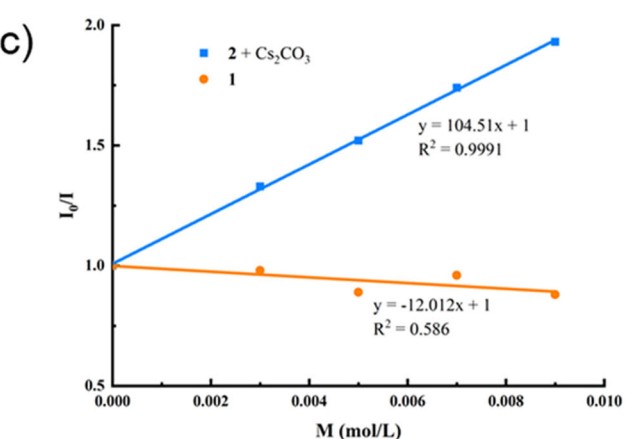

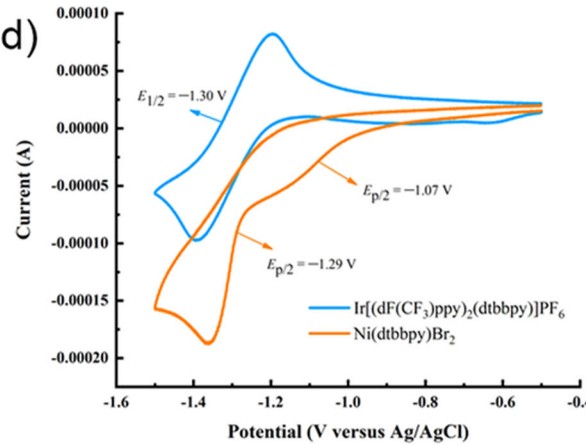

**Fig. 6 | Mechanistic experiments. a** The cyclic voltammogram of the carboxylic acid anion of *N*-propionylglycine (**2**) versus SCE in DMSO at 0.1 V/s scan rate. **b** Quenching of the photocatalyst Ir[dF(CF₃)ppy]₂(dtbbpy)PF₆ (5 × 10⁻⁵ M in DMSO) in the presence of increasing amounts of **2**. **c** Stern-Volmer fluorescence quenching experiments. **d** The cyclic voltammogram of Ir[(dF(CF₃)ppy)₂(dtbbpy)]PF₆ and Ni(dtbbpy)Br₂ versus Ag/AgCl in MeCN at 0.2 V/s scan rate.

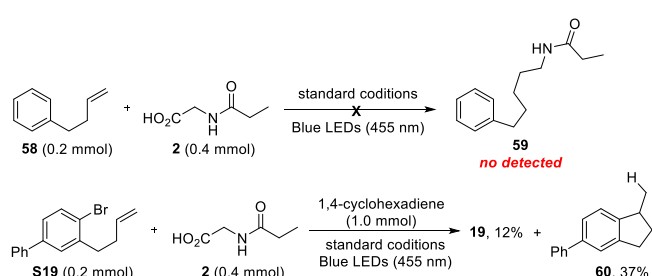

**Fig. 7 | Controlled experiments for mechanistic investigation.** Confirmation of the sequence of radical addition and cyclization events.

glove box. The mixture was stirred at room temperature for 30 min. Amino acid derivatives (0.30 mmol, 1.5 equiv.), MgCl₂ (4.8 mg, 0.05 mmol, 25 mol%), Cs₂CO₃ (108 mg, 0.33 mmol, 1.65 equiv.), 4CzIPN (3.2 mg, 0.004 mmol, 2 mol%) and the corresponding unactivated alkenes (0.20 mmol, 1.0 equiv.) were then added in sequence under the argon. The resulting mixture was then sealed and wrapped with electrical tape and removed from the glove box. The reaction mixture was irradiated with two parallel 18 W LEDs

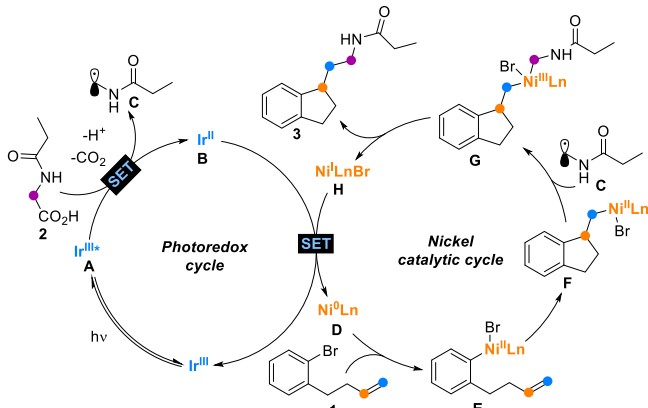

**Fig. 8 | Proposed mechanism.** Nickel/photoredox dual catalyzed arylalkylation of alkene.

(455 nm,) from a distance of approximate 8 cm for 48 h. The reaction was maintained at 40 °C by heating in an oil bath and cooling by a fan. Then, the solvent was evaporated and concentrated, the residue was purified by silica chromatography.

## Data availability

All data to support the conclusions are available in the main text or the Supplementary Information. All other data are available from the corresponding author upon request.

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

## Acknowledgements

Financial support for this work was provided by the National Natural Science Foundation of China (22271246 and 22061043), the Natural Science Foundation of Yunnan Province (202201AT070068), and Yunling Scholar Project and Young Talent Project of "Yunnan Revitalization Talent Support Program".

## Author contributions

C.X. and Z.P. conceptualized the project, directed the project, and finalized the manuscript draft. Y.G. conducted the optimization of the dual catalytic dicarbofunctionalization and part of the scope investigation. L.G., E.Z., Y.Y., M.J. and J.Z. performed scope investigation. All authors contributed to discussions.

## Competing interests

The authors declare no competing interests.
