## [Peer Review File · Nature Communications]

REVIEWER COMMENTS

Reviewer #1 (Remarks to the Author):

Xia and Pan reported a dual nickel- and photoredox-catalyzed dicarbonfunctionalization of alkenes to construct cyclization products. Although the author emphasized that this is a rare conversion of unactivated alkenes, the intramolecular design reduces the novelty of this work. Even so, the reviewer still believe that this is a nice advance in this field and might be suitable for publication on Nature Communications after addressing the following issues.

1 The most important point of this work is to afford a method for the cyclization compounds bearing an imine group. The author should make more efforts on the application scope to give a clear understanding for the readers. Now, only 5- and 6-membered carbocyclic frameworks can be generated. How about 7-membered cycles? How about the O- or N-linked cycles? At this moment, the scope seems too narrow and too special.

2 Identically, only mono-substituted alkenes were tested. How about internal alkenes, 1,1-disubstituted alkenes?

3 It is not easy to construct the chiral center in this product. The reviewer feels confused why the authors only give several examples in the asymmetric pattern. More examples including product 39 - 41 should be given to illustrate the scope.

4 The unsuccessful substrates should be given, at least in SI, to give more insights for the chemists who want to use this method.

5 the retention time of racemic and chiral sample 22 was too much different.

Reviewer #2 (Remarks to the Author):

The manuscript by Professor Xia and co-workers entitled "Nickel/Photoredox Dual Catalyzed Arylalkylation of Nonactivated Alkenes" presents a method to dual arylalkylation of nonactivated alkenes using nickel/photoredox dual catalysis system. This reaction follows α -Amino acids and dipeptides serve as starting materials, providing a wide range of indole amine derivatives through photo oxidative reduction decarboxylation. A few examples of enantioselective dicarbofunctionalization were also given. The utility of this dual catalysis system was demonstrated by synthesis of two pharmaceutically active compounds. However, some more work is required to

complete this manuscript. The manuscript can be supported for publication in Nat. Commun. after major revision.

Comments:

- 1)The following references should also be added. For nickel/photoredox catalytic dicarbofunctionalization of nonactivated alkenes reactions: Chem. Sci. DOI: 10.1039/d3sc01945d.; Chem. Sci., 2020,11, 4131-4137.
- 2)Line 113 – Selective functionalization of additional nonactivated olefin attached to the aryl ring did not interfere with the dicarbofunctionalization is an interesting and useful result. A discussion of this outcome based on the difficulty and easy in cyclization of the two groups should be added.
- 3)Although the mechanism has been studied from CV and fluorescence quenching experiments, the author should demonstrate the process of migration and insertion of inactive olefins after the oxidative addition of Ni⁰ with bromobenzene, so as to exclude the addition of olefins by free radicals generated from amino acid decarboxylation and then cyclization with Ni^{II}. For example, the protonated intramolecular migratory insertion intermediate products of Ni^{II} will be strong evidence of this process (J. Am. Chem. Soc. 2019, 141, 18, 7637–7643).
- 4)Line 378 – The corresponding references of 30 cannot be retrieved and this partment should be thoroughly proofread. A SET event between the Ni^I intermediate (E_{1/2}red [Ni^{II}/Ni⁰]) are not accurate according to photoredox cycle mechanization, E_{1/2}red [Ni^I/Ni⁰] should be confirmed by corresponding CV experiments or supported by relevant literature (Chem. Sci. DOI: 10.1039/d3sc01945d).
- 5)Line 220 –The resolution on the figure of mechanistic experiments and the NMR spectra and are poor.
- 6)Supplementary Information – Many of the provided NMR spectra indicate that the compound is not pure, e.g S8 on page S64, S10 on page S68, S12 on page S71, S18 page S77...etc.
- 7)Supplementary Information – several of the integrations in the NMR spectra are not approximately whole numbers, which is needed to reprocess, e.g 14 on page S106, 17 on page S109, 18 on page S110, 19 on page S111... etc. I have not listed them all.
- 8)Supplementary Information – the figure of reaction structure is not fully displayed in the diagram on page 17.

Reviewer #1 (Remarks to the Author):

Xia and Pan reported a dual nickel- and photoredox-catalyzed dicarbonfunctionalization of alkenes to construct cyclization products. Although the author emphasized that this is a rare conversion of unactivated alkenes, the intramolecular design reduces the novelty of this work. Even so, the reviewer still believe that this is a nice advance in this field and might be suitable for publication on Nature Communications after addressing the following issues.

Reply: We are grateful to the reviewer for supporting our work.

1 The most important point of this work is to afford a method for the cyclization compounds bearing an imine group. The author should make more efforts on the application scope to give a clear understanding for the readers. Now, only 5- and 6-membered carbocyclic frameworks can be generated. How about 7-membered cycles? How about the O- or N-linked cycles? At this moment, the scope seems too narrow and too special.

Reply: Thank you for your valuable suggestions. We ran a series of experiments along those lines. It was discovered that the O- and N-linked substrates were also applicable in the dual-catalyzed dicarbonfunctionalization. Moreover, the N-linked substrate (compound 29) demonstrated higher yields than the O-linked substrate (compound 28). We also examined the reaction in assembling 7-membered cycle. It was found that no product was generated in the C-linked substrate. However, moderate yield was obtained when N-linked substrate was subjected to the reaction conditions (compound 31). We added the following text to the revised manuscript: "Besides the C-linked substrates, the O- and N-linked substrates were then subjected for the dicarbonfunctionalization. It was found that the N-linked product (29) was harvested in much higher yield than the O-linked product (28). Meanwhile, we also tried to probe whether this dual-catalyzed strategy was applicable for generation of larger membered products. However, no 7-membered product was detected for the C-linked substrate. Instead, when O-linked substrates were exploited, the corresponding 7-membered cyclization compound 31 was isolated, albeit in low as 25% yield."

2 Identically, only mono-substituted alkenes were tested. How about internal alkenes, 1,1-disubstituted alkenes?

*Reply: Thank you very much for your nice comments and suggestions. We conducted the internal alkenes (both the 1,2-disubstituted and the 1,2,2-trisubstituted) and found this method was not applicable. The 2,2-disubstituted alkenes were also examined and delivered the products in moderate to good yields. We added the following text to the revised manuscript: “Finally, the disubstituted terminal alkenes delivered products in moderate to good yields (**32** and **33**), while the internal alkenes failed.”*

3 It is not easy to construct the chiral center in this product. The reviewer feels confused why the authors only give several examples in the asymmetric pattern. More examples including product 39 - 41 should be given to illustrate the scope.

*Reply: We fully agree with the referee and performed six more different examples, including the products **39** – **41**, to illustrate the scope of enantioselective arylalkylation. We rewrote the text in the revised manuscript as: “A total of twelve compounds were illustrated in the asymmetric version with good enantioselectivities. In comparison to the electron-withdrawing substituents ((*S*)-**9** and (*S*)-**19**), the electron-donating substituents ((*S*)-**4** and (*S*)-**18**) proved to be more efficient in both yield and enantioselective protocol. More sterically indanethylamine (*S*)-**22** was also delivered in 46% yield and 96:4 er. Various acyl groups were well appropriate*

so that dicarbofunctionalized products were provided in good yields and high ee values ((*S*)-**3**, (*S*)-**37**, (*S*)-**39**, and (*S*)-**41**, 45%–85% yield, 93:7–96:4 er). Other amino acids (such as alanine, leucine and lysine) with additional substituents on α -position were then evaluated for the enantioselective dicarbofunctionalizations. The tertiary stereogenic carbon displayed good enantioselectivity, while poor selectivity was observed on the amino acid moiety ((*S*)-**45**–(*S*)-**47**).”

4 The unsuccessful substrates should be given, at least in SI, to give more insights for the chemists who want to use this method.

Reply: Thank you very much for your nice comments and suggestions. We have added the unsuccessful examples into the revised SI as Supplementary Note 12.

Supplementary Note 12. Unsuccessful examples.

5 the retention time of racemic and chiral sample 22 was too much different.

Reply: Thank you for the nice comments. We wrongly uploaded the HPLC of racemic sample with different eluting system as that of chiral sample during the preparation of SI. We corrected this mistake with the following data.

No.	Retention Time min	Area mAU*s	Height mAU	Relative Area %
1	17.975	5.31626e4	1012.82910	50.7132
2	20.410	5.16674e4	1138.24170	49.2868

No.	Retention Time min	Area mAU*s	Height mAU	Relative Area %
1	17.838	2064.49365	52.58988	4.0988
2	20.213	4.83040e4	1078.38696	95.9012

Reviewer #2 (Remarks to the Author):

The manuscript by Professor Xia and co-workers entitled "Nickel/Photoredox Dual Catalyzed Arylalkylation of Nonactivated Alkenes" presents a method to dual arylalkylation of nonactivated alkenes using nickel/photoredox dual catalysis system. This reaction follows α -Amino acids and dipeptides serve as starting materials, providing a wide range of indole amine derivatives through photo oxidative reduction decarboxylation. A few examples of enantioselective dicarbofunctionalization were also given. The utility of this dual catalysis system was demonstrated by synthesis of two pharmaceutically active compounds. However, some more work is required to complete this manuscript. The manuscript can be supported for publication in Nat. Commun. after major revision.

Reply: We are grateful to the reviewer for supporting our work.

Comments:

1)The following references should also be added. For nickel/photoredox catalytic dicarbofunctionalization of nonactivated alkenes reactions: Chem. Sci. DOI: 10.1039/d3sc01945d.; Chem. Sci., 2020,11, 4131-4137.

Reply: Thank you very much for your nice comments and suggestions. We have cited the Molander and Wang works in the revised manuscript as new refs. 33 and 34 in the main text.

2)Line 113 – Selective functionalization of additional nonactivated olefin attached to the aryl ring did not interfere with the dicarbofunctionalization is an interesting and useful result. A discussion of this outcome based on the difficulty and easy in cyclization of the two groups should be added.

Reply: Thank you for your valuable suggestions. We added the following text to the revised manuscript: "The formation of compound 18 as a single product suggested that the reaction was initiated from the nickel-catalyzed intermolecular cyclization instead of the radical addition to alkene."

3)Although the mechanism has been studied from CV and fluorescence quenching experiments, the author should demonstrate the process of migration and insertion of inactive olefins after the oxidative addition of Ni⁰ with bromobenzene, so as to exclude the addition of olefins by free radicals generated from amino acid decarboxylation and then cyclization with Ni^{II}. For example, the protonated intramolecular migratory insertion intermediate products of Ni^{II} will be strong evidence of this process (J. Am. Chem. Soc. 2019, 141, 18, 7637–7643).

Reply: Thank you very much for your nice comments and suggestions. We conducted two controlled experiments for the mechanistic investigation. The first one is a substrate without bromide in the benzene ring. No radical coupling with alkene was detected from the reaction mixture, suggesting that the radical addition with unactivated alkene is no favorable. The second one is addition of 1,4-cyclohexadiene as protonation reagent. A protonated byproduct 60 was isolated in 37% yield, confirming the reaction was initiated with the oxidative addition

of Ni⁰ with bromobenzene. We added the following text and scheme to the revised manuscript: “To exclude the addition of olefins by free radicals generated from amino acid decarboxylation and then cyclization with Ni^{II}, two control experiments were conducted (Scheme 6). The reaction of **58** with **2** gave no cross-product, suggesting that a radical addition of the unactivated alkene is not favored. The selective coupling of radical with the two alkenes in compound **18** also confirms the reaction sequence (Scheme 2). Moreover, in the presence of 1,4-cyclohexadiene, the reaction of **S19** with **2** gave both the desired product **19** and protonated byproduct **60**.⁵⁴ These results support the reaction pathway that involves the activation of Ar–Br with Ni⁰, intramolecular migratory insertion, and alkyl radical coupling.”

Scheme 6. Controlled experiments for mechanistic investigation.

4)Line 378 – The corresponding references of 30 cannot be retrieved and this partment should be thoroughly proofread.

Reply: Thank you very much for your nice comments. We re-checked the mentioned reference 30 (Zuo, Z., Ahneman, D.T., Chu, L., Terrett, J.A., Doyle, A.G. & MacMillan, D.W.C. Merging Photoredox with Nickel Catalysis: Coupling of α -Carboxyl sp^3 -Carbons with Aryl Halides. *Science* **345**, 437-440 (2014)) and confirmed its journal name, publication year, volume number and page number are correct. We also proofread all other references for their correctness.

Science Current Issue First release papers Archive About

HOME > SCIENCE > VOL. 345, NO. 6195 > MERGING PHOTOREDOX WITH NICKEL CATALYSIS: COUPLING OF α -CARBOXYL sp^3 -...

REPORT f t in g w e

Merging photoredox with nickel catalysis: Coupling of α -carboxyl sp^3 -carbons with aryl halides

ZHIWEI ZUO, DEREK T. AHNEMAN, LINLING CHU, JACK A. TERRETT, ABIGAIL G. DOYLE, AND DAVID W. C. MACMILLAN Authors Info & Affiliations

SCIENCE • 5 Jun 2014 • Vol 345, Issue 6195 • pp. 437-440 • DOI: 10.1126/science.1255525

16,375 1,152 🔔 📌 🗣️ 📄

A SET event between the NiI intermediate ($E_{1/2\text{red}}[\text{NiII}/\text{NiI}]$) are not accurate according to photoredox cycle mechanization, $E_{1/2\text{red}}[\text{NiI}/\text{Ni}^0]$ should be confirmed by corresponding CV experiments or supported by relevant literature (Chem. Sci. DOI: 10.1039/d3sc01945d).

Reply: Thank you very much for your nice comments and suggestions. We measured the $E_{p/2}[\text{Ni}^{\text{I}}/\text{Ni}^0]$ as suggested as -1.29 V versus Ag/AgCl in MeCN. The $E_{p/2}[\text{Ir}^{\text{III}}/\text{Ir}^{\text{II}}]$ was also measured in MeCN as -1.30 V versus Ag/AgCl. The data supported the SET event between NiI/Ni0 and $\text{Ir}^{\text{III}}/\text{Ir}^{\text{II}}$ for the catalytic cycles. The following figure was added to the revised manuscript as figure 1d (the original figure 1d for light-on-off experiment was moved to SI). We also cited Abigail Doyle (JACS 2016, **138**, 12719; JACS 2021, **143**, 15873), Cristina Nevado (JACS 2023, **145**, 12532), and Chuan Wang's (CS, 2023, **14**, 6449) results to support the proposed SET process.

d) The cyclic voltammogram of $\text{Ir}[(\text{dF}(\text{CF}_3)\text{ppy})_2(\text{dtbbpy})]\text{PF}_6$ and $\text{Ni}(\text{dtbbpy})\text{Br}_2$ versus Ag/AgCl in MeCN at 0.2 V/s scan rate.

5) Line 220 – The resolution on the figure of mechanistic experiments and the NMR spectra and are poor.

Reply: Thank you very much for your kind suggestions. We have replaced the figures with a higher resolution figure.

6) Supplementary Information – Many of the provided NMR spectra indicate that the compound is not pure, e.g S8 on page S64, S10 on page S68, S12 on page S71, S18 page S77...etc.

Reply: Thank you very much for your nice comments. We re-purified the following compounds with impurities and recorded their NMR spectra.

7)Supplementary Information – several of the integrations in the NMR spectra are not approximately whole numbers, which is needed to reprocess, e.g 14 on page S106, 17 on page S109, 18 on page S110, 19 on page S111... etc. I have not listed them all.

Reply: Thank you very much for your nice comments. We have been reprocessed the integrations for the following compounds and revised the Supplementary Information.

8)Supplementary Information – the figure of reaction structure is not fully displayed in the diagram on page 17.

Reply: *Thank you very much for your nice comments. We replace the figure on page 17 with a full-displayed one.*

REVIEWERS' COMMENTS

Reviewer #1 (Remarks to the Author):

The authors have addressed the comments from the reviewers. It is acceptable for publication.

Reviewer #2 (Remarks to the Author):

The authors have carefully and fully revised the manuscript according to the opinions of the reviewers, and the revised manuscript can be accepted and published.

Reviewer #1 (Remarks to the Author):

The authors have addressed the comments from the reviewers. It is acceptable for publication.

Reply: We are grateful to the reviewer for supporting our work. We also highly appreciate your efforts in review.

Reviewer #2 (Remarks to the Author):

The authors have carefully and fully revised the manuscript according to the opinions of the reviewers, and the revised manuscript can be accepted and published.

Reply: We are grateful to the reviewer for supporting our work. We also highly appreciate your efforts in review.